# Effectiveness of online training in improving primary care doctors' competency in brief tobacco interventions: A cluster-randomized controlled trial of WHO modules in Delta State, Nigeria

Nnamdi Stephen Moeteke[1,2¤*], Patrick Oyibo[3,4‡], Oboratare Ochei[1,4‡], Maureen Iru Ntaji[1,4], Nyemike Simeon Awunor[1,4], Mitchell Oritsewino Adeyemi[1], Ibobo Mike Enemuwe[1,4], Eseoghene Agbatutu[1], Oluwaseun Opeyemi Adesoye[1]

1 Department of Community Medicine, Delta State University Teaching Hospital, Oghara, Delta State, Nigeria, 2 Center for Primary Care, Department of Global Health and Social Medicine, Harvard Medical School, Boston, Massachusetts, United States of America, 3 Department of Health Services Research and Management, City, University of London, London, United Kingdom, 4 Department of Community Medicine, Delta State University, Abraka, Delta State, Nigeria

¤ Current address: Department of Community and Public Health, Idaho State University, Pocatello, Idaho, United States of America

‡ PO and OO contributed equally to this work and are joint second authors.

* nnamdimoeteke@gmail.com

## Abstract

### Background

The World Health Organization (WHO) strongly recommends that brief tobacco interventions should be routinely offered in primary care. However, medical doctors do not consistently and effectively intervene during their encounters with cigarette smokers. There is a paucity of studies assessing the effect of training on the tobacco intervention competency of primary care doctors in Nigeria.

### Aim

To evaluate the effectiveness of online training in improving competency in brief tobacco interventions among primary care doctors in Delta State, Nigeria.

### Methods

A cluster-randomized controlled trial was conducted among eligible doctors working in government-owned facilities. The 22 eligible Local Government Areas (LGAs) served as clusters. The intervention group received a WHO six-hour online course on brief tobacco cessation intervention, delivered via Zoom. The control group received no intervention. A structured questionnaire was sent to participants via WhatsApp before and six months after the training. The primary outcome variables were scores for knowledge, attitude, self-efficacy, and practice. Differences in change of scores between intervention and control groups

**Data Availability Statement:** All relevant data are within the manuscript and its Supporting Information files.

**Funding:** The authors received no specific funding for this work.

**Competing interests:** The authors have declared that no competing interests exist.

were assessed with *t*-test. To adjust for clustering, these inter-group differences were further analyzed using linear mixed-effects regression modeling with study condition modeled as a fixed effect, and LGA of practice entered as a random effect.

## Results

The intervention group had a significantly higher mean of change in scores for knowledge (effect size 0.344) and confidence (effect size 0.52).

## Conclusion

The study shows that training, even online, positively affects clinician competency in brief tobacco intervention. This is important for primary care systems in developing countries. Mandatory in-service training and promotion of the WHO modules are recommended.

## Introduction

There are over a billion smokers in the world, 80% of whom live in low- and middle-income countries (LMICs) [1, 2]. It is projected that tobacco use will cause one in six deaths among adults (close to ten million deaths each year) by 2030, 80% of which will occur in LMICs [1, 2]. The critical role of smoking cessation in decreasing the burden of tobacco-related death and disability has been increasingly evident in recent years [3]. Clinical modalities for tobacco cessation and treatment of dependence are either behavioral or pharmacological [4]. Behavioral therapy entails practical counseling (in problem-solving/ skills training/ stress management), providing social support as part of treatment, and helping to secure social support outside of treatment [4]. **Brief behavioral cessation interventions** refer to cessation counseling delivered regularly during routine clinic consultations whether or not patients are seeking help with smoking cessation. They last less than 10 minutes [5]. **Intensive interventions** are more comprehensive and individualized, offered by Tobacco Treatment Specialists through individual/ group counseling, cessation support groups, and telephone counseling in the setting of addiction treatment/mental health centers or social service agencies [1]. Intensive interventions are more effective but more time- and cost-demanding, and, therefore, far less available, particularly in developing countries [1, 6]. Better results are obtained when counseling and medications are combined. Both should be used, except where pharmacotherapy is contraindicated [4]. There is no significant difference between the costs of cigarettes and Nicotine Replacement Therapy [7]. Routine brief counseling and low-cost drugs (such as Nortriptyline and Cystisine) are affordable for and in countries at all levels of economic development [8]. Treatment of tobacco dependence is "the 'gold standard' in preventive medicine and health care cost-effectiveness" [9]. Article 14 of the World Health Organization (WHO)'s Framework Convention on Tobacco Control (FCTC)—"the world's first public health treaty"—mandates countries to elaborately promote cessation and draw up treatment guidelines to be made widely available along with treatment services [10].

Because of its comprehensive, whole-person focus, experts have identified the primary care setting as ideal for the integration of screening for tobacco use and offering brief cessation intervention to tobacco users at every contact [11]. One strategy that has been proposed for the treatment of tobacco use in primary care is the 5As/5Rs Model of brief intervention (Fig 1) which requires less than 10 minutes of direct clinician time [12]. The critical nature of the

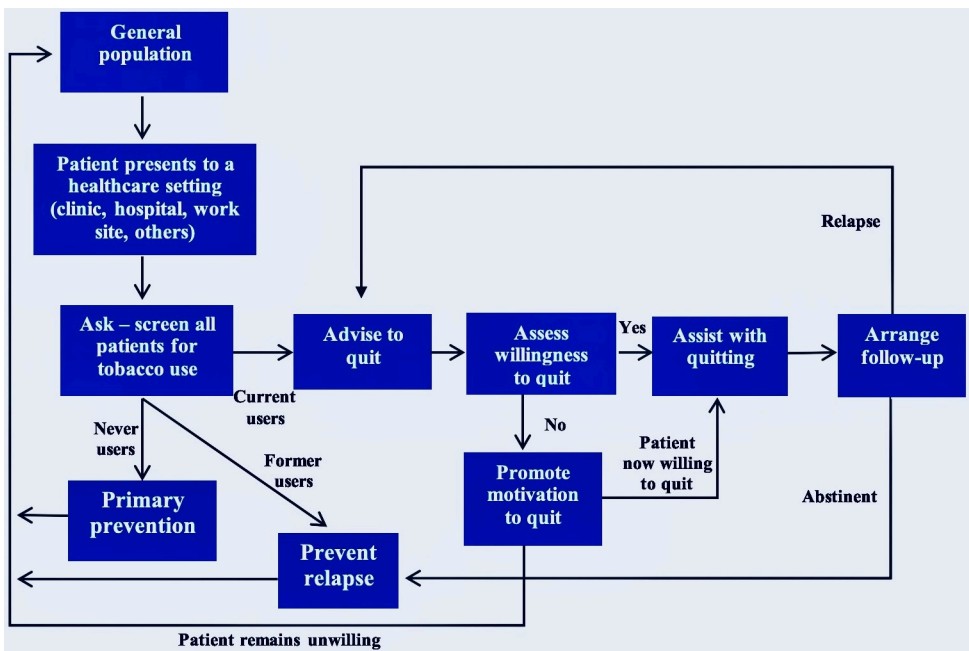

**Fig 1. The 5As/5Rs model for treatment of tobacco use and dependence.**

doctor-patient relationship and the enormous confidence reposed on doctors means that the place of the primary care doctor in cessation intervention is paramount [3, 8]. Smokers rate physician advice highly, and "physician-delivered interventions such as counseling are strongly supported by a solid body of evidence" [9]. Ironically, physicians do not consistently and effectively intervene when they come in contact with smokers, except if the immediate ailment is tobacco-related [8]. They usually believe that the patients are not interested in the advice and that intervening is outside their primary duties; physicians, themselves, may even be poorly informed about tobacco cessation [8]. Other barriers to the delivery of cessation intervention include fear of hurting patients' feelings, perceived ineffectiveness of cessation advice, deficiency of counseling skills, low provider self-efficacy, time constraints, etc. [13].

Despite the unequivocal stance of the WHO, as of 2018, only six middle-income countries and one low-income country offered comprehensive cessation support in primary care [14]. The establishment of effective cessation services has been hindered by an absence of capacity building for healthcare providers, competing disease priorities (e.g. HIV and TB), and a lack of data and locally applicable tobacco cessation guidelines [15]. Nigeria currently has neither a national tobacco cessation strategy nor clinical guidelines, and there is virtually no form of training for healthcare workers before and after graduation [14]. Nearly 70% of current smokers in the country would like to quit [14]. However, a nationally representative survey of Nigerian physicians showed that less than a third had good knowledge of smoking cessation [16]. Over 70% reported that tobacco education in the medical school curriculum is inadequate, and about two-thirds cited poor knowledge of the issue as the greatest obstacle to their implementing cessation intervention [16]. There is a poor perception of and low capacity to provide smoking cessation services among healthcare workers in Nigeria, with only 5% of respondents in a study reporting having ever received such training [17]. Another study reported poor implementation of the 5A's by physicians, with many missed opportunities for cessation intervention [18].

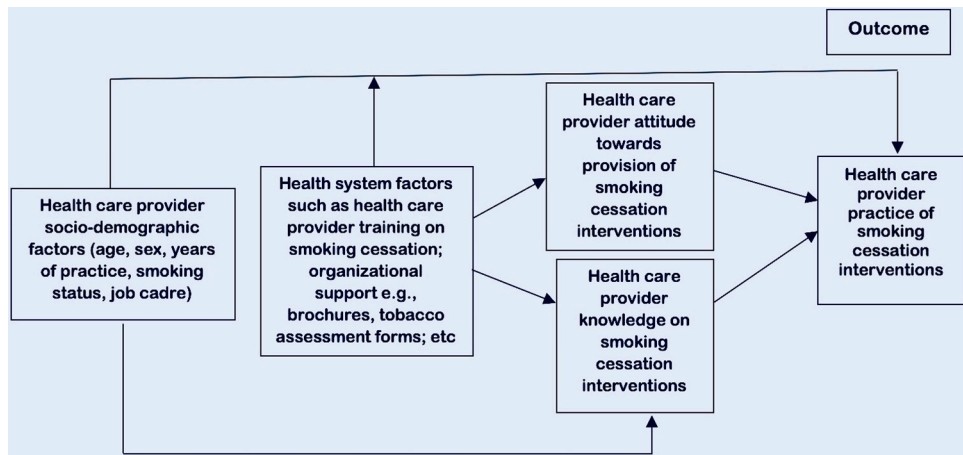

**Fig 2. Conceptual framework for the predictors of health care providers' competency in smoking cessation interventions.**

*Competency* is the sum of knowledge, attitude, and ability to perform a given task that eventually translates to behavior that enables a professional to perform their role effectively [19]. Various studies have highlighted the contributory factors in the performance of brief cessation interventions by healthcare personnel (Fig 2) [5, 20–22]. The 'behavior change theories' applied in health promotion, each with its own explanatory variables, can also be used to understand changes in the clinical practices of healthcare professionals [23]. One such theory is the Social Learning Theory (SLT). The three key explanatory factors in SLT (Fig 3), which must be modified for a new behavior to be acquired, performed, and maintained, are *behavioral capacity*, *efficacy expectations*, and *outcome expectations*. *Behavioral capacity* refers to the requisite knowledge and skills for the performance of the desired behavior. *Efficacy*

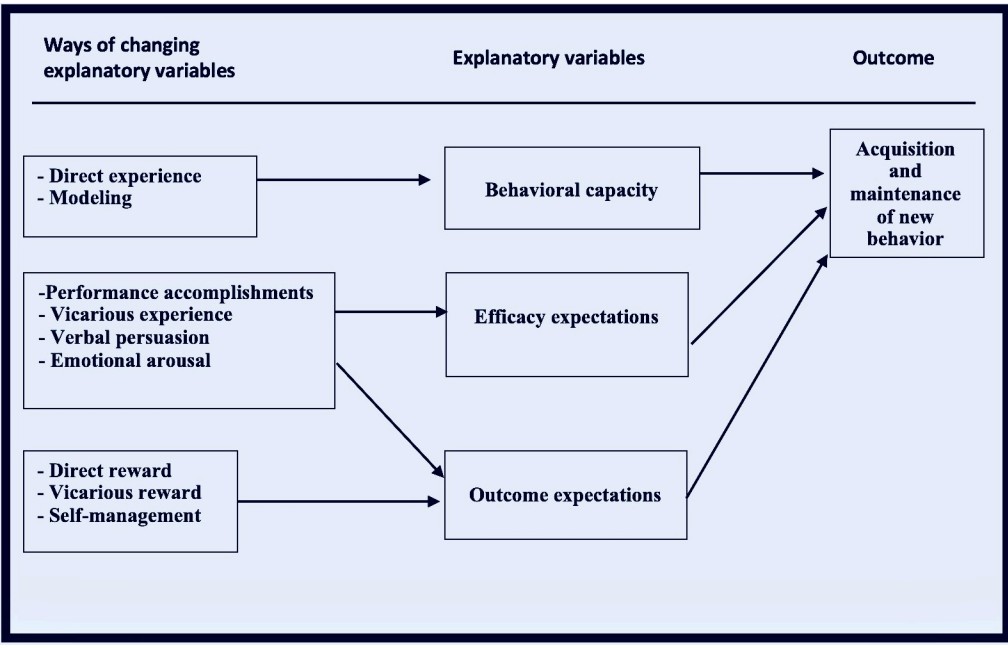

**Fig 3. Explanatory variables and processes of the social learning theory.**

*expectations* (*Self-efficacy or Confidence*) are the beliefs regarding one's ability to successfully execute a desired behavior. Perceptions of self-efficacy are based on people's assessment of themselves–whether they have the knowledge and skills to make changes in their behavior and whether external factors such as time and money will allow that change [24]. *Outcome expectations* are the beliefs that the performance of a particular behavior will lead to desired effects or consequences. SLT as a framework can be applied in the evaluation of an educational intervention [25]. In an assessment of an intervention to train primary care physicians on brief tobacco cessation, it would be important to measure the change in the practice of the desired behavior, as well as change in the explanatory variables: knowledge/skill levels, perceived self-efficacy, and beliefs about the usefulness of the behavior. The desired behavior would be the consistent practice of the 5A's model, and *Outcome expectations* (attitudes) would be the degree to which primary care physicians expect that carrying out the 5A's model among their patients who smoke will lead to cessation [24].

There is a paucity of studies assessing the effect of training on the smoking cessation competency of primary care physicians in Nigeria. A synthesis of literature from other regions indicates that training health professionals in smoking cessation is effective [26–33]. However, most available studies fall short of appropriate design, sampling technique, and sample size. The necessity of adapting training programs to varied cultures and evaluating their effectiveness in different settings has been highlighted [8, 26]. Given the constraints in low-income countries like Nigeria, online courses may be more desirable because they "allow distance learning when local training opportunities and resources are scarce" [26]. There is a need to fill the knowledge gaps and provide evidence to inform policy on the development of workforce competency. Therefore, this study aimed to evaluate the effectiveness of online training with WHO modules in improving competency in brief tobacco cessation interventions among primary care doctors in Delta State, Nigeria.

## Materials and methods

### Study area

This study was conducted in Delta State, which is in the south-south geo-political zone of Nigeria. The population of the State is 4,112,445 (males: 2,069,309; females: 2,043,136) [34]. Delta State has 25 Local Government Areas (LGAs) distributed into 3 senatorial districts. Eight of these LGAs are urban. A smoking prevalence as high as 15.8% has been reported for the general population [35]. Primary medical and dental care is delivered in the General Outpatient Clinics of primary, secondary, and tertiary health facilities by Medical/Dental Officers (doctors without specialist training) and resident doctors/consultants in Family Medicine or Community Medicine. There are 318 government-employed primary care doctors in the state.

### Study design

A cluster-randomized controlled trial (CRCT) design was necessitated by the nature of the intervention, to avoid "contamination" in the control group [36]. Local Government Areas (LGAs) served as clusters and the units of randomization.

### Study population

The study population consisted of primary care doctors working in government-owned health facilities.

## Selection/Eligibility criteria

Primary care doctors who had practiced for at least one year post-registration were included. Those who were scheduled to proceed on study leave or had chronic illnesses requiring recurrent days off duty were excluded to reduce the chances of inability to participate in the training and follow-up assessment. Those working in two or more clinics in different LGAs were also excluded (to prevent contamination). LGAs with less than four doctors were excluded based on the calculation for the required minimum size for each cluster (shown below).

## Sample size determination

The minimum sample size per group for a cluster-randomized controlled trial, $N_C = N_I \times$ VIF, where $N_I$ is the minimum sample size for an individual-randomized controlled trial, and VIF is the Variance Inflation Factor [37].

VIF is needed because of the likely similarities between participants within a cluster which makes it inappropriate to handle them as purely independent observations. Sample sizes for cluster-randomized trials should be larger to maintain statistical power [38]. The increment is given by the *design effect* or *variance inflation factor (VIF)*: $1 + (m-1)\rho$, where $m$ is the average cluster size, and $\rho$ is the intra-cluster correlation coefficient [39].

The minimum sample size per study arm for an individual-randomized controlled trial ($N_I$) is calculated using the following formula [38].

$$N_I = \frac{2 \times (Z_{\alpha/2} + Z_\beta)^2 \times S^2}{\delta^2}$$

Where

S = Pooled standard deviation of both comparison groups

$\delta$ = Anticipated difference between the group means ($\mu_1 - \mu_2$)

$Z_{\alpha/2} = Z_{0.05/2} = Z_{0.025} = 1.96$ (from Z table) at type-1 error of 5%

$Z_\beta = Z_{0.20} = 0.842$ (from Z table) at a power of 80%

Therefore,

$$N_I = \frac{2 \times (1.96 + 0.842)^2 \times S^2}{(\mu_1 - \mu_2)^2}$$

$$N_I = \frac{2 \times (1.96 + 0.842)^2}{(\mu_1 - \mu_2)^2 / S^2}$$

Since $(\mu_1 - \mu_2)/S = $ *Effect Size*, a standardized quantitative measure of the magnitude of an experimental effect [40],

$$N_I = \frac{2 \times (1.96 + 0.842)^2}{Effect\ Size^2}$$

$$N_I = \frac{15.7}{Effect\ Size^2}$$

In a previous educational intervention study on tobacco cessation counseling among health professionals [32], the *effect size* was 0.52.

Therefore, $N_I = 15.7 \div 0.52^2 \approx 59$ per arm

Singh *et al.* have determined the intra-cluster correlation for a number of process-of-care outcome measures at the primary care practice level and estimated the value for tobacco dependence treatment to be 0.054 [41].

As such, the VIF for this study was $1 + (12.56 - 1)0.054 = 1.61128$, and the minimum sample size per arm for the CRCT, $N_C = N_I \times$ VIF $= 59 \times 1.61128 \approx 96$.

To take care of possible attrition this was increased by 10% to 106 per arm (212 in total).

## Sampling, randomization, and recruitment

The minimum number of clusters required per arm when there are unequal cluster sizes is

$$k = \{N_I [1 + ((cv^2 + 1)\ m - 1)\ \rho\} \div m$$

where *cv* represents the coefficient of variation of the cluster sizes [37].

$k = \{60[1 + ((1.288949^2 + 1)12.56-1)0.054\} \div 12.56 = 8.62 \approx 9$

The required minimum size for each cluster is given by the formula below [37].

$N_I (1- \rho) \div [K—N_I (cv^2 + 1) \rho]$, where K is the number of clusters available.

$= 60(1–0.054) \div [25–60(1.288949^2 + 1)0.054] = 3.47 \approx 4$

Given that there were three LGAs with less than four government-employed primary care doctors out of the 25 LGAs in Delta State, all eligible primary care doctors in all the 22 eligible LGAs were recruited for the study.

Using the official list of personnel from the relevant government agencies, recruitment took place between 21st March and 2nd April 2022.

Stratified randomization was employed. The eligible clusters were divided into three sub-groups based on senatorial districts. Each subgroup was further stratified into rural and urban, and clusters in each sub-stratum were randomly allocated to each study arm by simple randomization using a random number table.

## Intervention

The intervention was the WHO e-learning course *Training for primary care providers*: *brief tobacco interventions* [42]. "WHO encourages primary care providers worldwide to attend this training to improve their knowledge, skills, and confidence to assist tobacco users in quitting as part of routine practice in primary care" [42]. The course (which consists of six modules) is self-paced and takes an average of six hours to complete. Three of the authors (who are public health physicians with WHO certification in brief tobacco interventions) guided participants through the course in a Zoom meeting tutorial. A PowerPoint presentation obtained from the WHO website was used [43]. This presentation covers the six modules and three more (Table 1). Participants were free to choose from one of five standardized training sessions held on five different days, between 22nd April and 3rd May 2022. Each session lasted for six hours in total, apart from two fifteen-minute breaks. The control group did not receive any training until after the study.

## Study instrument

This study employed a self-administered structured questionnaire adapted from previously validated instruments by Gichuki [5]. The questionnaire was pretested among twenty primary care medical practitioners in a neighboring state and revised for clarity. The questionnaire was sent to participants electronically via WhatsApp before the training and six months after. It is made up of five sections developed to assess socio-demographic independent variables (age, sex, smoking status and years of practice, job cadre, etc.), and dependent variables (knowledge, attitude, level of confidence, and practice of smoking cessation intervention, based on the 5A's model).

## Method of data collection

Data was collected in two phases viz pre-training and six months post-training.

The questionnaire section on **"Knowledge"** consisted of 9 questions and 11 statements that were used to evaluate the level of awareness of the participants as regards the benefits of providing smoking cessation interventions, behavioral smoking cessation intervention methods/techniques, pharmacological smoking cessation interventions, and nicotine dependence and withdrawal symptoms. Knowledge was assessed using a 3-point scale: "Yes"/"No"/ "Don't know" for the questions, and "True"/ "False"/ "Don't know" for the statements.

The section on **"Attitude"** comprised 9 statements that were used to assess the mindset of the participants with respect to the role of health care providers in the provision of smoking cessation interventions, the effect of smoking cessation interventions on the health care

**Table 1. Overview and learning outcomes of the training modules.**

| Overview and time taken to complete | Learning outcomes |
|---|---|
| **Module 1: The role of primary care providers in tobacco control and tobacco dependence treatment (40 minutes)** • The role of health professionals as role models, clinicians, educator, scientist, leader, opinion-builder, alliance-builder, and watching out for tobacco industry activities <br>• Description of brief advice used in the WHO FCTC Article 14 guidelines <br>• Brief tobacco intervention as an opportunistic intervention <br>• Effectiveness of brief advice on quitting | **Participants should be able to** • Acknowledge their roles in tobacco control and tobacco dependence treatment <br>• Describe the purpose of this training course <br>• Describe existing effective tobacco dependence treatment methods <br>• Describe the definition, effectiveness, feasibility, and content of brief tobacco interventions |
| **Module 2: Basics of tobacco use and tobacco dependence (40 minutes)** • Overview of local, national, and worldwide patterns of tobacco use. <br>• Theory and evidence on the three elements of tobacco addiction: physical/physiological dependence; emotional/psychological connection; and habitual and social connection | • Identify patterns of tobacco use (local, national, international) <br>• Describe the health, social and economic impact of tobacco use on tobacco users and others <br>• Clarify common misconceptions held by tobacco uysers <br>• Explain the benefits of quitting tobacco use <br>• Describe why people smoke and why they don't stop. |
| **Module 3: Overview of brief tobacco interventions (40 minutes)** • The primary purpose of a brief tobacco intervention <br>• The population impact of a brief tobacco intervention <br>• The several structured brief tobacco intervention models that can guide primary care providers through the right process to talk to patients about tobacco use and deliver advice (5A's/5R's) | • Describe the purpose and population impact of a brief tobacco intervention <br>• Describe at least three brief tobacco intervention models |
| **Module 4: Asking, advising, and assessing readiness to quit (40 minutes)** • How to ask about tobacco use and document tobacco use status in the medical record <br>• Theory of why advice should be personalized and how to tailor advice for a particular patient <br>• Theories of motivation (when is someone ready to quit?) and how to assess readiness to quit | • Ask and advise patients about their tobacco use in an appropriate way <br>• Use two ways to assess patients' readiness to quit |
| **Module 5: Dealing with low motivation (40 minutes)** • Overview of the 5R's approach and where it should be inserted during a brief intervention <br>• Use of some motivational tools to motivate patients for quitting tobacco use such as cost calculators; photos of smoking-exacerbated facial ageing; the carbon monoxide (CO) monitor; risk charts. | • Describe the 5R's brief tobacco intervention model <br>• Respond appropriately to exhibited stop-smoking resistance, employing the 5R's model <br>• Respond appropriately in cases of low motivation to quit, using motivational tools |
| **Module 6: Assisting and arranging for follow-up (40 minutes)** • Actions that can be taken to aid patients in quitting: helping to develop a quit plan; providing practical counseling to deal with challenges; providing intra-treatment social support; recommending pharmacotherapy if appropriate; providing supplementary materials <br>• Arranging for follow-up contacts <br>• Review of each stage of the 5A's and 5R's models | • Assist patients to stop tobacco use by helping them with a quit plan and providing intra-treatment social support and supplementary materials <br>• Arrange follow-up contacts <br>• Arrange a referral to specialist services if available <br>• Deliver a full, brief tobacco intervention according to the 5A's and 5R's models |
| **Module 7: Addressing Non-smokers' exposure to Second-hand Smoke (40 minutes)** • Overview of second-hand smoke <br>• Use of the 5A's model to offer a brief intervention to educate non-smokers about the dangers of second-hand smoke and advise them on avoiding the effects of second-hand smoke | • Describe the definition and dangers of second-hand smoke <br>• Describe the brief intervention model for reducing non-smokers' exposure to second-hand smoke |

**Table 1.** (Continued)

| Overview and time taken to complete | Learning outcomes |
|---|---|
| **Module 8: Pharmacotherapy (40 minutes)** • Nicotine Replacement Therapy (nicotine gum, trans-dermal patch, lozenge, oral inhaler, and nasal spray), bupropion and varenicline:<br>• what those medications are<br>• the purpose of using those medications<br>• available dosage<br>• advantages and disadvantages<br>• who can use those medications<br>• general guidelines for using those medications<br>• side-effects and warnings<br> • Methods of assessing the level of nicotine dependence | • Describe effective tobacco cessation medications<br>• Prescribe the available range of NRT products<br>• Recommend bupropion and varenicline<br>• Apply tools to assess tobacco users' levels of nicotine dependence. |
| **Module 9: Promoting brief tobacco interventions in the community (40 minutes)**<br> • Community resources for delivering brief interventions: tobacco quit lines, specialist services in cessation clinics, local tobacco cessation classes and support groups, smokers' web-based assistance, free self-help materials | • Identify outreach opportunities for delivery of brief tobacco interventions to tobacco users in their homes or community settings<br> • Identify referral resources within a local community for the primary care provider to deliver brief tobacco interventions |

provider-patient relationship, acceptance of smoking cessation interventions by patients, and availability of time to provide smoking cessation interventions during routine consultations. Each response was scored on a 3-point scale ("Agree", "Indifferent", "Disagree").

**Self-efficacy (confidence)** of the participants in providing smoking cessation interventions was measured with 8 questions based on the 5A's strategy and the participants were required to rate their confidence in performing the various activities using a 3-point scale: "not at all confident", "a little confident" and "fully confident".

In the section on **"Practice",** ten questions were used to assess, using a 3-point scale, the frequency of performance of different procedures in the 5A's model in daily interactions with patients ("Never", "Sometimes", and "Always").

## Data management

a) Measurement of outcome variables

Total scores for knowledge, attitude, confidence, and practice were calculated for each participant as follows.

Each correct response to the knowledge-based questions or statements attracted a score of 1, and each incorrect/ "don't know" response, was zero. The highest possible total score was 20. A score of 1 was given for each response agreeing with an appropriate attitude-based statement or disagreeing with an inappropriate one, while a score of zero was given for each response disagreeing with an appropriate statement, agreeing with an inappropriate one, or being indifferent. The highest possible total score was 9.

For confidence-based questions, a score of zero was awarded for "not confident", one for "a little confident" and two for "confident". Total confidence scores could range from 0 to 16.

For practice-based questionnaire items, a score of two was awarded for "always", one for "sometimes" and zero for "never". Possible total scores could range from 0 to 20.

b) Statistical analyses

The collected data was entered into a Microsoft Excel spreadsheet and exported to IBM SPSS Statistics v. 25 for analysis. Continuous variables were summarized as 'mean ± standard

deviation' while categorical variables were summarized as frequencies and percentages. The independent variables were socio-demographic and other characteristics as well as study condition ('received training' or 'no training'). The outcome variables were levels of knowledge, attitude, self-efficacy, and practice for each participant expressed as total scores. The changes in the outcome measures (post-training) were also calculated.

Bivariate analyses included an initial baseline comparison of variables between study arms using Chi-square and $t$-test to assess comparability and the success of randomization. Participants who were lost to follow-up were also compared with those who completed the study in each arm. Analysis of differences between intervention and control groups for change of total scores of the outcome variables was done with $t$-test. To allow for clustering by LGA, these inter-group differences were further analyzed using linear mixed-effects regression modelling with 'study condition' modelled as a fixed effect and 'LGA of practice' entered as a random effect [44]. The model was also adjusted for potential confounders noted at the baseline comparison of the study arms. Bonferroni correction for multiple comparisons was included. All outcome analyses were by intention-to-treat, according to the pre-intervention random allocation. The level of significance for all analyses was set at $\alpha = 0.05$.

## Ethical considerations

The Institutional Review Board of Harvard University determined that this study is exempt (Protocol number IRB22-0197) while the Health Research Ethics Committee of the Delta State University Teaching Hospital, Oghara gave ethical approval (Approval Number: HREC/PAN/2021/008/0420). The participants signed the written informed consent form after reading the Participant Information Document and getting answers to any questions they had (from the investigator or the ethics committee). Participants were assured of their confidentiality. It was made clear that they could freely decline participation at any point during the research.

Since the online course required hours of internet connection, allowing participants to bear the cost would have contravened the principle of non-maleficence. An appropriate data bundle (commensurate with the duration of the online training on Zoom), five CME points, and a certificate of training were given to each participant. These were provided as compensation and a token of appreciation which are generally acceptable "because they aim to make a participant whole, are quite minimal or are offered in a way that would not influence decisions to participate" [45].

There are ethical concerns about withholding a potentially efficacious intervention from participants in the control arm of a study. In the absence of an intervention in current use, the control arm may receive no intervention, provided they have identical access to all the treatment and preventive services as those in the intervention arm [38]. This study sought to assess the efficacy of an online training package in improving physicians' competency in smoking cessation for which there is yet no modality already in use in Nigeria. In line with the principle of beneficence, at the end of the study, participants in the control arm were offered the same training as those in the intervention arm.

## Results

The pre-intervention response rates were 88.8% (91.8% in the intervention arm, and 85.5% in the control arm). The total number of subjects at the end of the study was 261 (141 in the intervention group, and 120 in the control group) (S1 File). This gave post-intervention response rates of 88.7% in the intervention arm and 82.8% in the control arm (Fig 4).

At baseline, the study and control groups were only significantly different in age, years of medical practice, and level of health care to which their facility belonged (Table 2). Concerning

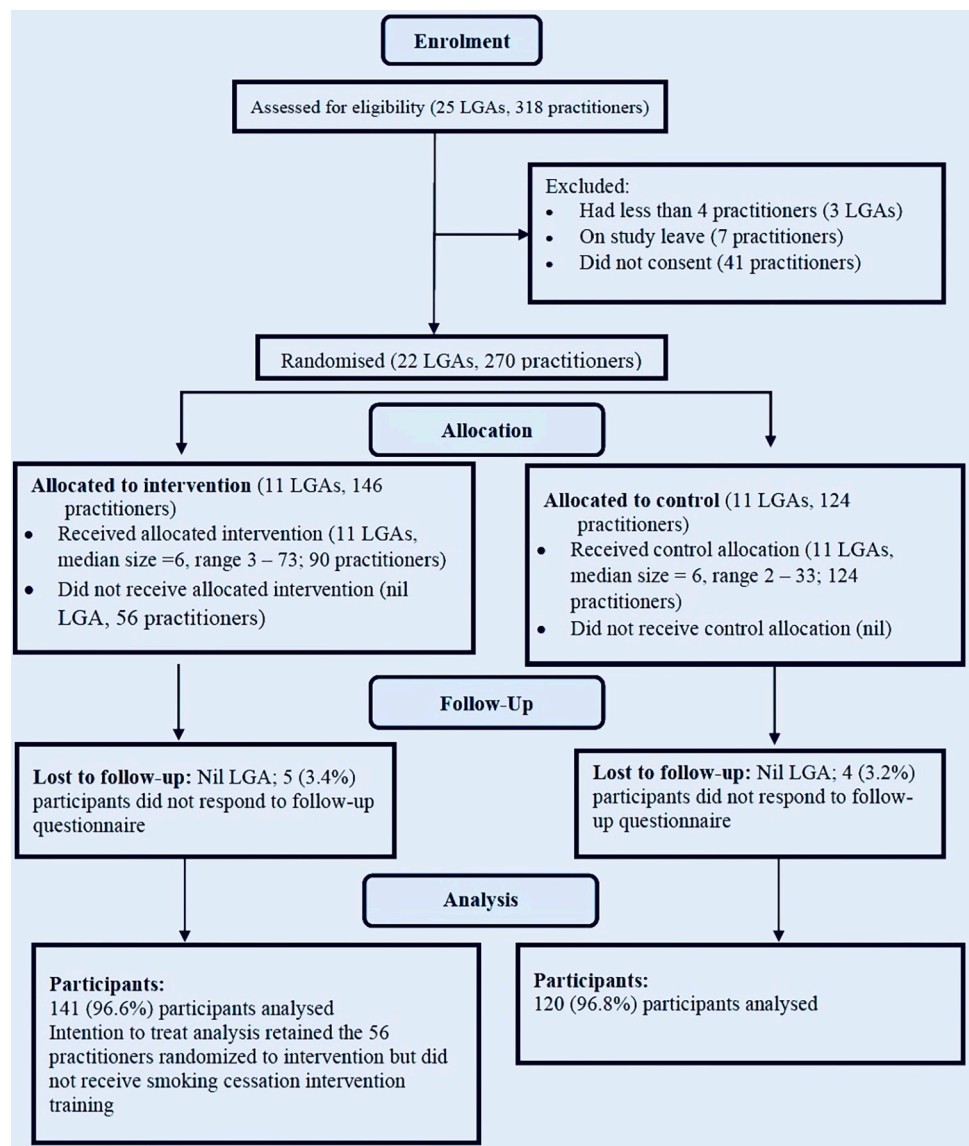

**Fig 4. The flow of clusters and participants through the trial.**

sociodemographic characteristics and work-related variables, there was no statistically significant difference between participants who were lost to follow-up and those who completed the study. The intervention group had a significantly higher mean of change in score for knowledge, confidence, and practice (Table 3). However, after adjusting for clustering and possible confounders, the difference was only significant for knowledge and confidence (Table 4).

## Discussion

Six months post-intervention, there was a significantly higher improvement in the knowledge and confidence of participants in the intervention group compared to those in the control group. This is similar to the results of a randomized controlled trial (RCT) conducted among community-based health professionals in Scotland which showed a significant improvement in knowledge, and cluster-randomized controlled trials (CRCT) in Switzerland and the United

**Table 2. Baseline socio-demographic characteristics of participants.**

| | Control (n = 124) | | Intervention (n = 146) | | Total (N = 270) | | Test statistic | p-value |
|---|---|---|---|---|---|---|---|---|
| **Age (years)** | | | | | | | | |
| Mean ± SD | 45.63±8.11 | | 43.08±7.34 | | 44.25±7.79 | | $t$ = 2.715 | **0.007** |
| **Years of experience** | | | | | | | | |
| Mean ± SD | 17.44±8.12 | | 14.88±7.22 | | 16.06±7.74 | | $t$ = 2.746 | **0.006** |
| **Sex** | *n* | % | *n* | % | *n* | % | | |
| Male | 89 | 71.8 | 91 | 62.3 | 180 | 66.7 | $\chi^2$ = 2.692 | 0.101 |
| Female | 35 | 28.2 | 55 | 37.7 | 90 | 33.3 | | |
| **Job description** | *n* | % | *n* | % | *n* | % | | |
| Consultant | 11 | 8.9 | 17 | 11.6 | 28 | 10.4 | $\chi^2$ = 6.856 | 0.077 |
| R. Doctor | 30 | 24.2 | 53 | 36.3 | 83 | 30.7 | | |
| M. Officer | 79 | 63.7 | 70 | 47.9 | 149 | 55.2 | | |
| D. Officer | 4 | 3.2 | 6 | 4.2 | 10 | 3.7 | | |
| **Level of healthcare of facility** | *n* | % | *n* | % | *n* | % | | |
| Primary | 14 | 11.3 | 15 | 10.1 | 29 | 10.8 | $\chi^2$ = 7.583 | **0.023** |
| Secondary | 77 | 62.1 | 69 | 47.3 | 146 | 54.1 | | |
| Tertiary | 33 | 26.6 | 62 | 42.6 | 95 | 35.1 | | |
| **Smoking status** | | | | | | | | |
| Current Smoker | 2 | 1.6 | 2 | 1.4 | 4 | 1.5 | $\chi^2$ = 0.027 | 0.987 |
| Ex-smoker | 11 | 8.9 | 13 | 8.9 | 24 | 8.9 | | |
| Never smoked | 111 | 89.5 | 131 | 89.7 | 242 | 89.6 | | |
| **Previous training on smoking cessation** | | | | | | | | |
| Yes | 25 | 20.2 | 25 | 17.1 | 50 | 18.5 | $\chi^2$ = 0.410 | 0.522 |
| No | 99 | 79.8 | 121 | 82.9 | 220 | 81.5 | | |

R–Resident; M–Medical; D–Dental | $\chi^2$ = Chi-square test | $t$ = Independent sample $t$-test

States which demonstrated an increase in confidence [46–48]. However, training had no significant effect on attitude and practice in this current study unlike RCTs done in Scotland and Indonesia, and CRCTs conducted in Finland, Switzerland, and the United States [32, 33, 46–49]. The observed ineffectiveness of the training in improving attitude in this current study may be due to the high level of good attitude pre-intervention. The observed difference in effect on practice could be because these previous studies adopted a workshop form of training which, in addition to educational methods such as lectures, PowerPoint presentations, and interactive sessions, included more participatory breakout role-play sessions with typical patient/case scenarios and demonstrations. This explanation is supported by SLT with its explanatory variables of behavior change and the processes through which these influences can be modified. SLT holds that *behavioral capacity* or *skill set* can be altered through

**Table 3. Comparison of means of the changes in total scores (95% confidence intervals) between the intervention and control groups at six months post-intervention.**

| Primary outcome | Intervention (n = 141) | Control (n = 120) | Difference between groups | t statistic (df) | p-value | Effect size |
|---|---|---|---|---|---|---|
| **Knowledge** | 1.61 (1.11–2.14) | 0.63 (0.26–1.00) | 0.985 (0.313–1.657) | 2.884 (259) | **0.0001** | 0.353 |
| **Attitude** | 0.20 (-0.08–0.44) | 0.29 (0.05–0.57) | -0.093 (-0.476–0.290) | -0.479 (259) | 0.870 | - |
| **Confidence** | 3.02 (2.41–3.65) | 0.75 (0.20–1.27) | 2.271 (1.470–3.073) | 5.578 (259) | **0.0001** | 0.656 |
| **Practice** | 0.91 (0.23–1.70) | 0.58 (0.001–1.20) | 0.332 (-0.654–1.317) | 0.662 (259) | **0.046** | 0.212 |

**Table 4. Adjusted comparison between groups for mean of the changes in total scores (95% confidence intervals) of outcome variables at six months\*.**

| Primary outcome | Intervention (n = 141) | Control (n = 120) | Difference between groups | t statistic (df) | p-value[a] | Effect size (95% CI) |
|---|---|---|---|---|---|---|
| Knowledge | 1.558 (0.923–2.193) | 0.599 (0.001–1.198) | 0.958 (0.092–1.825) | 2.456 (10.277) | **0.033** | 0.344 (0.099–0.589) |
| Attitude | 0.214 (-0.046–0.474) | 0.292 (0.011–0.573) | -0.077 (-0.460–0.306) | -0.398 (258) | 0.691 | - |
| Confidence | 2.748 (1.868–3.628) | 0.947 (0.080–1.815) | 1.800 (0.573–3.028) | 3.104 (16.324) | **0.007** | 0.520 (0.273–0.767) |
| Practice | 0.900 (0.228–1.572) | 0.583 (-0.142–1.309) | 0.317 (-0.672–1.306) | 0.631 (258) | 0.529 | - |

\*Adjusted for clustering (by LGA of practice) and group differences in age, years of practice, and level of health facility between groups using linear mixed-effects regression modeling

[a]Adjustment for multiple comparisons: Bonferroni

observational learning (watching someone else model the desired behavior) and participatory learning (direct experience through supervised practice and repetition). Also, important in SLT is the role of vicarious experience: observing peers successfully perform a behavior modifies one's expectations about their own behavior. Training intervention using participatory workshop modules achieves these through demonstrations and role plays. This current study, wherein training was done by online lectures and brainstorming/interactive sessions, lacked direct practice, successful individual performance of 5A's, and vicarious experience. Therefore, the failure of the intervention to improve practice could be tied to the absence of these modifiers [25]. Another explanation for the lack of improvement in practice can be deduced from this study's conceptual framework wherein practice is influenced firstly by the interaction of several health system factors (including the availability of organizational support such as brochures for patients, guidelines, specialist clinics for referral, financial coverage for cessation intervention) with knowledge and attitude [5, 20–22]. The absence of these factors in Nigeria may have made it impossible for the knowledge and confidence gained by the physicians from the training to translate into action. Despite the shortfall, the improvements in knowledge and confidence are useful. This is because the knowledge necessary for behavior change (behavioral capacity) is seen as a powerful tool in building self-efficacy. Self-efficacy has been proposed as the most important prerequisite for behavior change and will affect how much effort is put into a task and the outcome of that task. As such the development of knowledge and confidence are crucial to achieving behavior change [50]. Overall, this study supports the general suggestion in the wider literature that training, even online, has a positive effect on clinician competency, though actual change in behavior (increase in practice) is more likely if there is observational and participatory (practical) learning in keeping with SLT [24–33, 46–54].

A limitation of this study is the susceptibility of cluster trials to selection biases at the cluster level and the individual level (recruitment of individuals into the study) [55]. The former was minimized by ensuring careful randomization of clusters by an independent person. Full identification aided by the sampling frame (complete list of employees) from the relevant government agencies, and provision of compensation to participants in the training/study were used to maximize the inclusion of individuals within the clusters and reduce dropouts [55]. Some form of compensation and token of appreciation are generally acceptable in research, to the extent that they aim to make a participant whole, are minimal, and are offered in a way that does not influence the decision to participate [45]. Secondly, this study relied on self-reported data to assess the smoking cessation practice and confidence of participants. As such, some level of under/over-reporting may have occurred. The assurance of confidentiality and an appeal to the participants for candor was used to minimize this bias.

A strength of this study is the use of a previously validated instrument with established adequate Cronbach's alpha coefficients for the assessment of tobacco cessation intervention

knowledge, attitude, self-efficacy, and practice. Also, this study had a high response rate of 85.9% which has a positive implication for external validity. Individuals who turn down research invitations tend to be different from those who participate [56]. The extent of the non-response bias was determined by comparing respondents who completed the study with those who were lost to follow-up [44]. The small number of dropouts in each arm and the comparability with the complying population indicate that the bias is likely to be insignificant [36]. There were formal power considerations from the outset, and the sample size was also above the calculated required minimum which means that it was adequate to detect statistically significant differences where such exist [55]. Survey response rates have been on the decline in recent times due to progressively increasing time pressures on people, and medical doctors respond less than other health workers [55]. This threat was borne in mind especially with the shortage of medical personnel due to massive brain drain. Making repeated contact with selected study subjects is a strategy for improving recruitment [56]. This was employed with three courteous reminder messages (at roughly 72-hourly intervals) after the initial contact for each of the two phases of the study. This is one of the few studies on the effect of training on tobacco cessation competency that employed a linear mixed effects model to account for clustering. In addition, there was an adjustment for baseline factors and covariates which reduces the estimates of the standard errors and increases precision [36]. Adjusting for covariates can also improve the power of a trial by reducing the magnitude of the between-cluster variation [36]. These considerations make this study generalizable, and its findings useful for informing policy.

## Conclusion and recommendations

Using appropriate design, sampling technique, and sample size, this study's findings meet its aim to fill the gap in the literature in Nigeria regarding the effect of training on primary care doctors' tobacco intervention competency. To the best of our knowledge, this is the first study to provide data in this regard in Nigeria. Training improved knowledge and confidence scores, though there was no significant effect on attitude and practice. This study has important implications for primary care systems in Nigeria, and public health strategies for tobacco control through demand reduction measures. There should be formulation and implementation of policies that engender pre- and in-service training on cessation interventions for primary care doctors in Nigeria. The inclusion of smoking cessation in the curriculum of medical schools, and periodic continuing medical education courses, preferably in the form of in-person workshops, should be emphasized. However, online training can be adopted where resources or circumstances do not permit workshops. Information about the free WHO online course should be widely circulated, and its uptake encouraged. More research on improving competency in delivery cessation services among primary care doctors is needed. Studies to comparatively assess the effectiveness of online versus in-person training; determine if the length of training modulates effectiveness as well as the optimum duration; and examine the relative contributions of the various health system factors to the practice of cessation intervention will better inform policy [57]. Also, more studies in Nigeria that replicate this current one would be important to confirm its findings and build evidence for the generalizability of the observed effect.

## Supporting information

**S1 File. Minimal data set.**
(SAV)

## Acknowledgments

This study was conducted while Nnamdi Moeteke was a visiting scholar at the Harvard Medical School Center for Primary Care, under the Program in Global Primary Care and Social Change. We are grateful to the Center, and the Program's Director Dr David B. Duong for the opportunity, and their guidance and support.

## Author Contributions

**Conceptualization:** Nnamdi Stephen Moeteke, Patrick Oyibo, Oboratare Ochei.

**Formal analysis:** Nnamdi Stephen Moeteke.

**Investigation:** Nnamdi Stephen Moeteke, Mitchell Oritsewino Adeyemi, Ibobo Mike Enemuwe, Eseoghene Agbatutu, Oluwaseun Opeyemi Adesoye.

**Methodology:** Nnamdi Stephen Moeteke, Patrick Oyibo, Oboratare Ochei, Maureen Iru Ntaji, Nyemike Simeon Awunor, Mitchell Oritsewino Adeyemi, Ibobo Mike Enemuwe, Eseoghene Agbatutu, Oluwaseun Opeyemi Adesoye.

**Project administration:** Nnamdi Stephen Moeteke, Mitchell Oritsewino Adeyemi, Ibobo Mike Enemuwe, Eseoghene Agbatutu, Oluwaseun Opeyemi Adesoye.

**Resources:** Nnamdi Stephen Moeteke, Mitchell Oritsewino Adeyemi, Ibobo Mike Enemuwe, Eseoghene Agbatutu, Oluwaseun Opeyemi Adesoye.

**Supervision:** Patrick Oyibo, Oboratare Ochei, Maureen Iru Ntaji, Nyemike Simeon Awunor.

**Validation:** Nnamdi Stephen Moeteke, Patrick Oyibo, Oboratare Ochei, Maureen Iru Ntaji, Nyemike Simeon Awunor, Oluwaseun Opeyemi Adesoye.

**Visualization:** Nnamdi Stephen Moeteke, Patrick Oyibo, Oboratare Ochei.

**Writing – original draft:** Nnamdi Stephen Moeteke.

**Writing – review & editing:** Nnamdi Stephen Moeteke, Patrick Oyibo, Oboratare Ochei, Maureen Iru Ntaji, Nyemike Simeon Awunor, Mitchell Oritsewino Adeyemi, Ibobo Mike Enemuwe, Eseoghene Agbatutu, Oluwaseun Opeyemi Adesoye.

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
