## [Decision Letter · Decision Letter 0]

9 Jan 2024

PONE-D-23-28937Effectiveness of Online Training in Improving Primary Care Doctors’ Competency in Brief Tobacco Interventions: A Cluster Randomised Controlled Trial of WHO Modules in Delta State, NigeriaPLOS ONE

Dear Dr. Moeteke,

Thank you for submitting your manuscript to PLOS ONE. After careful consideration, we feel that it has merit but does not fully meet PLOS ONE’s publication criteria as it currently stands. Therefore, we invite you to submit a revised version of the manuscript that addresses the points raised during the review process.

We look forward to receiving your revised manuscript.

Kind regards,

Billy Morara Tsima, MD MSc

Academic Editor

PLOS ONE

4. Please upload a new copy of Figures 1 and 2 as the detail is not clear. Please follow the link for more information: https://blogs.plos.org/plos/2019/06/looking-good-tips-for-creating-your-plos-figures-graphics/" https://blogs.plos.org/plos/2019/06/looking-good-tips-for-creating-your-plos-figures-graphics/

5. Please include a new copy of Tables 1-4 in your manuscript; the current table is difficult to read. Please follow the link for more information: https://blogs.plos.org/plos/2019/06/looking-good-tips-for-creating-your-plos-figures-graphics/

Reviewers' comments:

Reviewer's Responses to Questions

**Comments to the Author**

1. Is the manuscript technically sound, and do the data support the conclusions?

Reviewer #1: Yes

Reviewer #2: Yes

2. Has the statistical analysis been performed appropriately and rigorously? 

Reviewer #1: Yes

Reviewer #2: Yes

3. Have the authors made all data underlying the findings in their manuscript fully available?

Reviewer #1: Yes

Reviewer #2: No

4. Is the manuscript presented in an intelligible fashion and written in standard English?

Reviewer #1: Yes

Reviewer #2: Yes

5. Review Comments to the Author

Reviewer #1: 1. Introduction:

- Introduction: well written. However, I am not sure whether the paragraph 4 "competency is the sum.... who smokes will lead to cessation.25" fit at that place. It is valuable information, but it does not fit in this paper where it is now. I suggest authors shorten it and fit it somewhere in paragraph 1 of the introduction, which is the background.

- aim of the study is well written.

2. MATERIALS AND METHODS:

Study setting: Are all doctors involved in the care of smokers? About how many doctors per cluster?

Selection/Eligibility Criteria: The authors mentioned that "those who were scheduled to proceed on in-service training...". The authors were conducting an evaluation of training level 2 (Kirkpatrick model of evaluation of training). Does this imply that the training was exclusively conducted through in-service lectures? If this is the case, it presents a significant drawback, as in-service lectures are designed to facilitate training that primarily focuses on the lower levels of knowledge and attitude in Bloom's taxonomy. Furthermore, it should be noted that in-service lecture training is not designed to achieve learning outcomes that specifically aim to bring about changes in high-level attitudes and behaviours. Tutorials and workshops may be employed to facilitate the acquisition of advanced levels of knowledge (Bloom's taxonomy) as well as foster the development of desired attitudes and behaviours. If these measures were put into effect, it would be necessary for authors to explicitly describe the level of interactivity involved in the training and provide a clear outline of how these interactions were organised. Additionally, it is worth considering whether authors implemented a blended learning approach or utilised the flipped classroom method of instruction. In summary, implementation of in-service training by authors would result in trainees being unable to acquire a high level of knowledge according to Bloom's taxonomy, thereby leading to no discernible change in high level of attitude or practise. Hence, what is the rationale for undertaking a study when the outcome is already anticipated?

I encourage authors to read about the types of training needed to accomplish learning outcomes, such as

-Butcher C, Davies C, Highton M. Designing Learning: From Module Outline to Effective Teaching [Internet]. 2nd ed. Second Edition. | New York: Routledge, 2020. |: Routledge; 2019 [cited 2023 Nov 17]. Available from: https://www.taylorfrancis.com/books/9780429873799. Read chapter 5 from page 70, most the table 5.1

3. RESULTS:

Well presented

4. DISCUSSIONS:

It was obvious to find that participants gained a low level of knowledge (knowledge, comprehension, and application) and attitude (aware) based on the type of training the authors implemented. Maybe they need to emphasise this as a major weakness of their study.

Reviewer #2: Summary of research: This manuscript describes a cluster randomized controlled trial, that evaluate the effectiveness of online training with WHO modules in improving competency in brief tobacco cessation interventions among primary care doctors in Delta State. The intervention resulted in significant improvement in knowledge and confidence of the primary care doctors,however, training did not significantly improve attitudes and practice of doctors.

Minor issues to be corrected

Figure 2:Conceptual framework illustrating the link between various demographic and health system factors with the health care providers’ knowledge, attitude and practice of smoking cessation intervention, distorted, could not review this figure.

6. PLOS authors have the option to publish the peer review history of their article (what does this mean?). If published, this will include your full peer review and any attached files.

Reviewer #1: **Yes: **Stephane Tshitenge

Reviewer #2: No

---

## [Author Response · Author response to Decision Letter 0]

15 Jan 2024

We have revised the manuscript to meet PLOS ONE’s style requirements, included new copies of Tables 1 – 4 in the manuscript, and uploaded new copies of Figures 1 and 2. We have also prepared a mark-up copy of the manuscript that highlights the changes made to the original version, as well as an unmarked version of the revised paper without tracked changes. Our data set is uploaded as a Supporting Information file. All corrections have been implemented. However, there are some exceptions and we would like to make clarifications on these.

Reviewer #1

Study setting: Are all doctors involved in the care of smokers

The World Health Organisation (WHO) Framework Convention on Tobacco Control (FCTC), other relevant documents, and experts have recommended that brief tobacco interventions be routinely offered in primary care and offered to smokers at every contact with a clinician, and that competency in this intervention is an expected standard of practice for every clinician. Our study population was primary care doctors, so they fall within the category of people who should be involved in tobacco intervention.

Selection/eligibility criteria: The authors mentioned that “those who were scheduled to proceed on in-service training…”

There appears to be some misunderstanding here with the use of “in-service training”. By this, we meant people who would be proceeding on study leave. So, we excluded them because of their likely unavailability for our intervention and data collection. Our intervention was not an “in-service training” in the sense that was elaborated by the reviewer. We appreciate the learning point and have used more appropriate terminology in that part of the manuscript.

---

## [Editor Report · Decision Letter 1]

1 Feb 2024

Effectiveness of online training in improving primary care doctors’ competency in brief tobacco interventions: A cluster randomised controlled trial of WHO modules in Delta State, Nigeria

PONE-D-23-28937R1

Dear Dr. Moeteke,

We’re pleased to inform you that your manuscript has been judged scientifically suitable for publication and will be formally accepted for publication once it meets all outstanding technical requirements.

Kind regards,

Billy Morara Tsima, MD MSc

Academic Editor

PLOS ONE
---

## [Editor Report · Acceptance letter]

13 Feb 2024

PONE-D-23-28937R1 

PLOS ONE

Dear Dr. Moeteke, 

I'm pleased to inform you that your manuscript has been deemed suitable for publication in PLOS ONE. Congratulations! Your manuscript is now being handed over to our production team.

Kind regards, 

on behalf of

Dr. Billy Morara Tsima 

Academic Editor

PLOS ONE